# Four Novel Disease-Causing Variants in the *NOTCH3* Gene in Russian Patients with CADASIL

**DOI:** 10.3390/genes14091715

**Published:** 2023-08-28

**Authors:** Fatima Bostanova, Polina Tsygankova, Ilya Nagornov, Elena Dadali, Lyudmila Bessonova, Aleksey Kulesh, Viktor Drobakha, Irina Danchenko, Ilya Kanivets, Ekaterina Zakharova

**Affiliations:** 1Research Centre for Medical Genetics, Moscow 115522, Russia; polgamma@yandex.ru (P.T.); nagornovilya@mail.ru (I.N.); genclinic@yandex.ru (E.D.); bessonovala@yandex.ru (L.B.); doctor.zakharova@gmail.com (E.Z.); 2Department of Neurology and Medical Genetics, Vagner Perm State Medical University, Perm 614990, Russia; aleksey.kulesh@gmail.com (A.K.); drobakha.v@gmail.com (V.D.); 3Perm Regional Clinical Hospital Perm Multiple Sclerosis Center, Perm 614015, Russia; irene-dan@mail.ru; 4Medical Center Genomed, Perm 614036, Russia; dr.kanivets@genomed.ru

**Keywords:** CADASIL, the *NOTCH3* gene, targeted gene sequencing, next-generation sequencing (NGS)

## Abstract

Background: Cerebral autosomal dominant arteriopathy with subcortical infarcts and leukoencephalopathy (CADASIL) is an inherited disease with unknown mechanisms and a broad phenotypic spectrum. It is caused by pathogenic variants in the *NOTCH3* gene. The symptoms of the disease mainly include recurrent strokes with vascular risk factors, migraine with aura, dementia, and mood disturbances. Case presentation: Peripheral blood samples were collected from five patients from four unrelated families to extract genomic DNA. In four patients, analysis of exons 2, 3, 4, 5, 6 and adjacent intronic regions of the *NOTCH3* gene was made via Sanger sequencing. Two previously undescribed nucleotide variants were identified in two patients: missense variant c.208G>T, (p.Gly70Cys) in exon 1 and splice-site variant c.341-1G>C in intron 3. Further DNA of two other patients were analyzed using a next-generation sequencing-based custom AmpliSeq™ panel for 59 genes associated with leukodystrophies. Two novel missense variants in the *NOTCH3* gene were identified, c.1136G>A, (p.Cys379Tyr) in exon 7 and c.1547G>A, (p.Cys516Tyr) in exon 10. The pathogenic variant c.1547G>A, (p.Cys516Tyr) was confirmed in the fifth patient (family case) by Sanger sequencing. All patients had a history of headaches, transient ischemic attacks, memory impairment, and characteristics of MRI results. Three patients had strokes and two patients had psychiatric symptoms. Conclusion: We found four previously undescribed pathogenic variants in the *NOTCH3* gene in five patients with CADASIL and described their clinical and genetic characteristics. These results expand the mutational spectrum of CADASIL.

## 1. Background

The most common form of hereditary cerebral angiopathy is cerebral autosomal dominant arteriopathy with subcortical infarcts and leukoencephalopathy (CADASIL). Typical clinical manifestations include migraine with aura, transient ischemic attacks (TIA), fixed focal neurological deficits due to lacunar infarcts, and cognitive decline [1]. The disease is caused by the heterozygous (de novo or inherited) pathogenic variants in the *NOTCH3* gene. The gene consists of 33 exons and localizes on chromosome 19q12. Exons 2 through 24 encode EGF-like domains (first identified in epidermal growth factor) with six cysteine residues. To date, most of the pathogenic variants responsible for the disease are located in these exons and change the highly conserved number of six cysteines in one of these EGFr domains to an uneven number of five or seven cysteines. A brain MRI shows leukoencephalopathy and lacunar infarcts.

Importantly, CADASIL is a progressive disease with insufficiently studied pathogenesis leading to disability. The diagnostic odyssey for most CADASIL patients is long and complex because of the lack of awareness about the disease in the medical community. Adding complexity is the symptomatic heterogeneity of the disease, frequently even between family members that carry the same mutation but develop different clinical features. Between 2007 and 2022, 72 cases of CADASIL were diagnosed at the Research Centre for Medical Genetics. Among them, four previously undescribed variants were identified. We aimed to study the spectrum of clinical manifestations in five patients with previously undescribed nucleotide variants in the *NOTCH3* gene and underline the importance of the detailed family history and careful evaluation of brain MRI data to optimize the diagnostic process.

## 2. Materials and Methods

### 2.1. Patients

Five patients (three females and two males) from four unrelated families were examined. The diagnosis of CADASIL syndrome was verified by the genealogical analysis, clinical picture, brain MRI, and the results of molecular and genetic analysis.

### 2.2. MRI Scan

An MRI brain examination was performed in T1 and T2 weighted modes (T1WI, T2WI) and fluid-attenuated inversion recovery (FLAIR) in three planes (axial, sagittal, and coronal). 

### 2.3. Molecular Genetic Methods

The isolation of genomic DNA was carried out from whole blood using the DNAEasy kit (QiaGen, Germantown, MD, USA), according to the manufacturer′s standard protocol.

Partial analysis of exons 2, 3, 4, 5, 6 and adjacent intron regions of the *NOTCH3* gene was carried out by automatic Sanger sequencing using ABIPrism 3500xl Genetic Analyzer (Applied Biosystems, Foster City, CA, USA), according to the manufacturer′s protocol. Primer sequences were designed using the reference sequence NM_000435.3.

The proband′s DNA (patients 3, 4.1) was analyzed using the original custom AmpliSeq™ panel on Ion S5 machine. The panel included coding exons of 59 genes which could be affected in different forms of inherited leukodystrophies. Sequencing results were analyzed using an in-home automatic algorithm for data analysis. Average coverage for the sample was 80×, with a coverage width of (20×) ≥ 90%. The detected variants were named according to HGVS nomenclature presented on the http://varnomen.hgvs.org/recommendations/DNA website (assessed on 24 July 2022).

To assess the population frequencies of the identified variants, we use data from 1000 Genome Projects, the ESP6500, and the Genome Aggregation Database v2.1.1 [2,3,4]. To assess the clinical significance of the identified variants, the OMIM database and the HGMD^®^ Professional version 20221.1 were used. An assessment of the pathogenicity and causality of genetic variants was carried out in line with the international recommendations for the interpretation of data obtained by massive parallel sequencing by ACMG and special *NOTCH3* consensus recommendations of the European Academy of Neurology [5]. All genetic variants are submitted to the ClinVar database (https://www.ncbi.nlm.nih.gov/clinvar/ (accessed on 21 January 2023)), accession numbers: VCV001879844.1, VCV001879845.1, VCV000994826.3.

### 2.4. Patients

#### 2.4.1. Patient 1

A 39-year-old woman consulted a geneticist for a diagnosis and prognosis for her offspring. Her mother (died at 69), aunt (died at 64), cousin (died at 40), grandfather (died at 47), and grandfather’s sister had an unspecified degenerative disease of the nervous system, accompanied by dementia, psychiatric disorders, and migraine (Figure 1). At age 39, the proband had an episode of slurred speech lasting 10 min. Examination by a neurologist revealed no focal neurological symptoms. Brain MRI revealed white matter hyperintensities (WHM) in the periventricular areas, which led to the suspicion of CADASIL syndrome. The c.208G>T, (p.Gly70Cys) substitution in exon 1 in the heterozygous state affecting a highly conserved protein region was identified by Sanger sequencing (Table 1). Since there were no living affected relatives, genetic testing for them was not possible. The clearly traceable autosomal dominant pattern of inheritance, debut at a young age, and similarity of clinical manifestations in relatives allowed us to confirm the diagnosis of CADASIL. The patient was referred under the care of a neurologist in the community. Within the subsequent two years, there were two more episodes of transient ischemic attack. Since the age of 43, headaches, dizziness, and memory impairment appeared. At the age of 49, an MRI of the brain revealed multiple white matter hyperintensities (WMH) lesions and multiple areas of cystic-gliosis changes; confluent areas of WMH were detected in symmetrical parts of the temporal lobes (Figure 1).

#### 2.4.2. Patient 2

The 45-year-old woman complained of periodic numbness of the left side of the trunk, labile systolic blood pressure, and anxiety attacks in a confined space [6]. The first symptoms started at the age of 22. She was hospitalized at the age of 38 with suspected acute cerebral circulation disorder due to speech impairment and weakness in the right extremities. She was examined by a neurologist at the Multiple Sclerosis Center where CADASIL was suspected. Multiple differently shaped (2–15 mm) irregular white matter hyperintensities (WMH) with clear contours on T2-weighted MRI images were identified. Vast, confluent areas of WMH were detected in symmetrical parts of the temporal lobes without the involvement of the cerebral cortex (Figure 2). The father had a “microstroke” at the age of approximately 50 and another with lethal outcome at the age of 59. According to the patient, the father had no headaches, migraines, or dementia. By partial analysis of the *NOTCH3* gene, the canonical splice-site substitution c.341-1G>C in the heterozygous state was detected (Table 1). In this case, bioinformatic analysis using numerous algorithms for predicting splicing disorders (SpliceAI, SPiP, MMSplice) demonstrates a high degree of influence of the c.341-1G>C variant on the splicing acceptor site of intron 3 of the *NOTCH3* gene, which disrupts its highly conserved dinucleotides. The peculiarity of clinical manifestations, specific changes on brain MRI, and the identification of a likely pathogenic variant in the *NOTCH3* gene allowed us to confirm the diagnosis to the patient. The family history revealed that the father had a “microstroke” at age 50 and another fatal stroke at age 59. According to the patient, the father had no headaches, migraines, or dementia. Unfortunately, in this case, it cannot be determined whether the variant was inherited or arose de novo.

#### 2.4.3. Patient 3

The 49-year-old woman complained of memory impairment, dizziness, concentration difficulties, and ataxia. She considered herself ill from the age of 45 after craniocerebral trauma (she fell down due to dizziness) after which she suffered from headaches. Later, an MRI of the brain (at the age of 48) showed signs of leukopathy. At the age of 49, the disease progressed with a sharp deterioration in the form of dizziness, nausea and vomiting, and with a suspected stroke. On examination, atactic gait and cognitive disturbances were noted. On the brain MRI, multiple white matter hyperintensities and subcortical lacunar lesions (Figure 3) were observed. Blood glucose concentration was elevated: 8.4 mmol/L (6–8.5 mmol/L in dynamics). From the family history, it is known that her sister (half-sibling by mother) died at the age of 48 from a stroke. She had the first stroke at the age of 37 and repeated multiple strokes afterwards. Her mother died at the age of 69 because of a stroke. The patient was referred by a neurologist for consultation at the Research Center for Medical Genetics. The presence of a history of stroke in the mother of the proband and the early age of stroke onset in the sister followed by rapid progression made it possible to suspect CADASIL. The nucleotide variant c.1136G>A, (p.Cys379Tyr) was detected in exon 7 of the *NOTCH3* gene in a heterozygous state by targeted panel sequencing (Table 1). Due to the absence of living family members with signs of the disease, it was not possible to assess the phenotype–genotype correlation.

#### 2.4.4. Patient 4.1

A 56-year-old man was sent to the Research Center for Medical Genetics for a consultation. He complained of memory impairment, fatigue, vision impairment, episodes of speech disturbances (while the patient himself did not realize them), unsteadiness of gait, and falling when walking downstairs. Fatigue, dizziness, speech disorders, and numbness in the left arm appeared for the first time at the age of 35. Self-relief occurred 30 min after rest. Similar attacks repeated twice within 1.5 months. According to the results of the brain MRI at the age of 48, the signs of lacunar strokes were found in the right frontal lobe, thalamus, putamen, and the pons, gliotic changes in the left hemisphere of the cerebellum, as well as multiple foci in the white matter. From the age of 47 to 56, there were approximately five episodes of disorientation, confusion, speech disorders, and dizziness. At the last hospitalization, a cerebral vascular accident (CVA) was established. In addition, at the age of 49, the patient was treated by a psychiatrist for depression and suicidal thoughts. Since the age of 56, he has not worked (as a dentist). During the examination, he showed foolishness, did not perceive his condition objectively, and did not follow the instructions (but sometimes made an effort). The speech was moderately dysarthric and reflexes were high from the hands and feet. In the complicated Romberg test, the patient was unstable. The proband’s son (Patient 4.2) complained of a periodic feeling of “heaviness” in his head, and an increase in blood pressure to 150/90 mmHg. At the age of 32, there was an attack of dizziness, disorientation, headache, and loss of speech. The attack lasted 10 min and then gradually stopped without special help. At the age of 33, two similar attacks occurred with a frequency of 1.5–2 weeks. The brain MRI showed white matter hyperintensities (Figure 4). On clinical examination, the geneticist showed no focal or neurologic symptoms. The targeted panel sequencing detected the nucleotide variant c.1547G>A (p.Cys516Tyr) in a heterozygous state in exon 10 of the *NOTCH3* gene in Patient 4.1 (Table 1). By Sanger sequencing, the pathogenic variant was confirmed in his son (Patient 4.2).

## 3. Discussion

HGMD^®^ (human gene mutation database) version 20221.1 contains totally 480 variants in the *NOTCH3* gene (missense—446, splicing—6, deletions—11, insertions—6, indels—7). Among 480 variants, 385 are described as deleterious (DD). ClinVar database (https://www.ncbi.nlm.nih.gov/clinvar/ (accessed on 21 January 2023)) contains 1279 variants in the *NOTCH3* gene (missense—664, nonsense—8, frameshift—22, splicing—5, deletions—38, insertions—29, indels—9, duplication—33). Among 1279 variants, 91 are described as likely pathogenic, 186 are pathogenic, and 424 are of uncertain significance. Almost all variants are unique for each family.

In the vast majority of cases, sequence changes in the *NOTCH3* gene are missense variants leading to the loss or gain of a cysteine residue in one of the EGF-like domains of the protein [7]. The relationship between the genotype and phenotype is currently unclear [7]. However, it was suggested that variants located in EGFr domains 1–6 were associated with more severe disease than variants in domains 7–34, including earlier age of stroke onset and shorter survival [8]. We analyzed the *NOTCH3* gene sequence in five patients with suspected CADASIL syndrome. Four novel pathogenic variants were identified in exons 1, 7, and 10 and in intron 3 of the *NOTCH3* gene, which are localized in EGFr domains 1, 9, and 13 (Table 1).

Recognition of CADASIL through early clinical features and imaging is a key to appropriate diagnosis and counseling [9]. Classic manifestations of CADASIL include recurrent ischemic stroke, transient ischemic attack, migraine, and cognitive impairment. Transient ischemic attacks and stroke are reported in approximately 85% of symptomatic individuals and mean age at onset of ischemic episodes is approximately 45–50 years [10,11]. Approximately 30% of CADASIL patients get affected by migraine attacks—the majority with aura—often as the first symptom of the disease [12]. Cognitive impairment is a common and relatively early presentation of CADASIL and often leads to dementia in a large number of patients. Neuropsychiatric manifestations of CADASIL include significant depression, anxiety, and apathy [13]. In larger cohort studies, it was found that 20–30% of patients experience major depression [9,14]. One of the studied patients (Patient 4.1) had manifestations of depression, which required psychiatric monitoring. Our other patient (Patient 2) complained of anxiety attacks in a confined space, which she had never experienced before. These clinical symptoms, which emphasize the psychiatric impact of progressive cerebral vascular dysfunction, are often debilitating elements of the clinical picture that can overshadow motor and cognitive symptoms [13]. 

In the diagnosis of CADASIL, instrumental findings outperform clinical findings. Typical manifestations on MRI include WMH and lacunar infarcts. WMH is a common and early MRI feature of CADASIL. Increased signal intensity in the anterior temporal lobe has high sensitivity and specificity in most cases [15]. This change was detected in two of our patients (patients 1, 2). Signs of recurrent cerebral microbleeds (cortical–subcortical regions, white matter, thalamus, pons) are identified in most patients with CADASIL and are usually localized outside ischemic foci, which allows for considering it almost a pathognomonic sign of the disease [16]. These signs were also present in our patients (patients 1, 2, 3, 4.1). Symmetric periventricular WMH, commonly affecting the anterior temporal lobe, can be observed long before the first symptoms appear [17,18]. By the age of 35, essentially all patients with CADASIL have abnormal MRI findings, which occasionally may occur in the absence of clear clinical features [19,20]. A detailed study of the family history plays an important role in the diagnosis of CADASIL. As stated in patient 1, the key to the diagnosis was a family history. Some patients with a molecularly confirmed diagnosis may remain asymptomatic for a long time, even with well-defined lesions on brain MRI. Brain MRI may be normal early in the disease course, but by the fifth decade, significant white matter changes are the rule rather than the exception [21]. In a study of a cohort of 301 CADASIL patients, it was found that higher WMH associated with the temporal lobes and frontal gyri correlated with a milder course of the disease as compared with WMH in the pyramidal tracts [22]. However, the question of correlation between the degree of WMH changes and the severity of CADASIL patients remains open today and requires further study.

The diagnosis of CADASIL can be confirmed by genetic testing with or without skin biopsy [23]. Electron microscopy shows characteristic granular osmiophilic material (GOM) within the vascular medium next to vascular smooth muscle cells. Detection of GOM is considered pathognomonic for CADASIL, but sensitivity has been reported to vary [24,25]. Skin biopsy is useful if the pathogenic variant is not detected or a nucleotide variant of unknown clinical significance in the *NOTCH3* gene is detected [26]. According to the ACMG/AMP variant interpretation guidelines, the terms “pathogenic variants” and “likely pathogenic variants” are synonymous in clinical settings, which means that both variants are considered diagnostic, and both can be used for clinical decision-making [5]. The nucleotide variants we identified in *NOTCH3* and the characteristic clinical and neuroimaging findings allowed us to establish the diagnosis of CADASIL without additional diagnostic studies (skin biopsy).

## 4. Conclusions

We have described four novel pathogenic variants in patients with CADASIL syndrome, which allowed us to expand the spectrum of causative pathogenic variants in the *NOTCH3* gene. The clinical diversity of this disease makes certain difficulties for timely diagnosis. Anamnestic, clinical, and neuroimaging data should be carefully compared to prevent misdiagnoses and optimize diagnostic search tactics.

## Figures and Tables

**Figure 1 genes-14-01715-f001:**
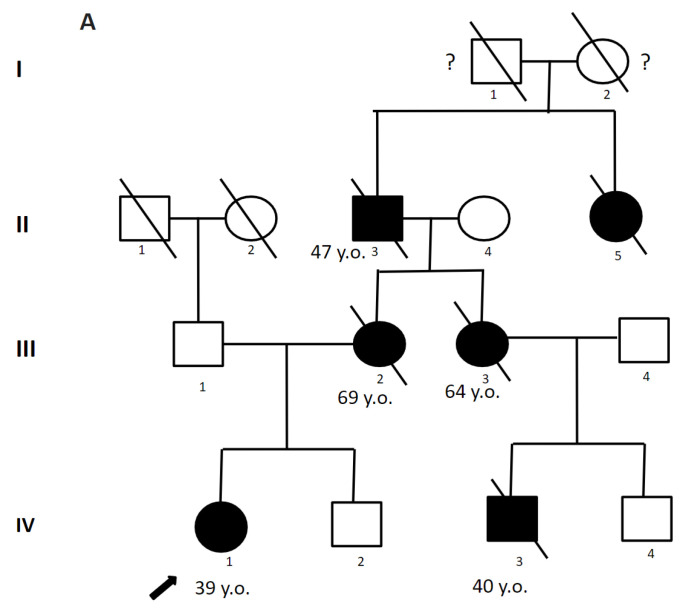
Pedigree and Neuroimaging features of patient 1. (**A**)—Pedigree of patient 1; (**B**)—MRI images of the brain (IV-1), red arrows show: (**a**–**c**,**e**,**f**)—multiple white matter hyperintensities and subcortical lacunar lesions; (**d**)—vast, confluent areas of WMH in symmetrical parts of the temporal lobes and subcortical lacunar lesion; (**e**)—cerebral atrophy.

**Figure 2 genes-14-01715-f002:**
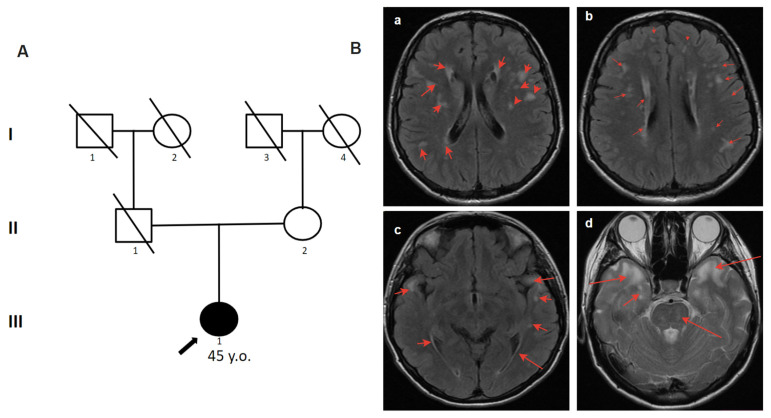
Pedigree and Neuroimaging features of patient 2. (**A**)—Pedigree of patient 2.; (**B**)—MRI images of the brain (III-1), red arrows show: (**a**,**b**)—multiple differently shaped (2–15 mm) irregular white matter hyperintensities (WMH) with clear contours on T2-weighted MRI images; (**c**,**d**)—vast, confluent areas of WMH in symmetrical parts of the temporal lobes without the involvement of the cerebral cortex.

**Figure 3 genes-14-01715-f003:**
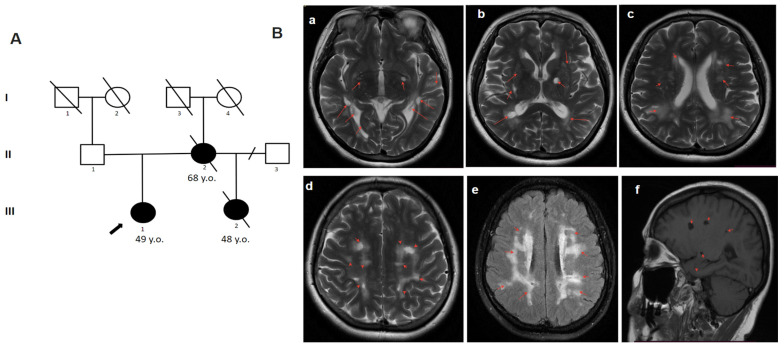
Pedigree and Neuroimaging features of patient 3. (**A**)—Pedigree of patient 3.; (**B**)—MRI images of the brain (III-1); (**a**–**f**)—multiple white matter hyperintensities and subcortical lacunar lesions (red arrows).

**Figure 4 genes-14-01715-f004:**
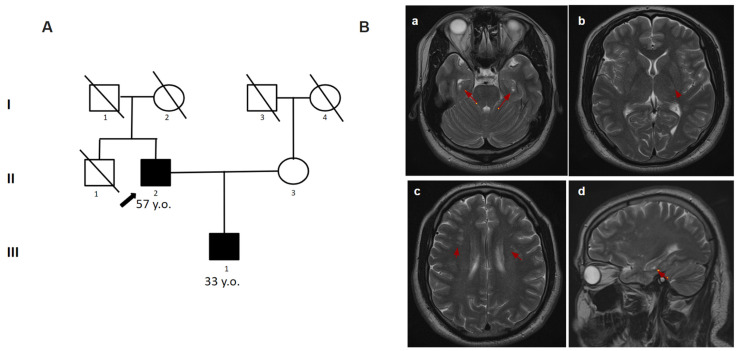
Pedigree and Neuroimaging features of patient 4.1 and 4.2. (**A**)—Pedigree of patient 4.1 and 4.2; (**B**)—MRI images of the brain patient 4.2 (III-1); (**a**–**d**)—multiple white matter hyperintensities (red arrows).

**Table 1 genes-14-01715-t001:** Four novel *NOTCH3* variants reported in the present study.

Patient Number	Family Number	Gender	Nucleotide Variants	Amino Acid Change	Exon/Intron Number	EGFr Domains	AGMG Classification	Evidence of Pathogenicity	Program
1	1	f.	c.208G>T	p.Gly70Cys	1 exon	1	Likely pathogenic	PM2, PP3, PM1, PP4	MutPred, DEOGEN2, M-CAP, Mutation assessor, MVP, FATHMM, FATHMM-MKL, PROVEAN
2	2	f.	c.341-1G>C	-	3 intron	-	Pathogenic	PM2, PVS1, PP3, PP4	SpliceAI, SPiP, MMSplice
3	3	f.	c.1136G>A	p.Cys379Tyr	7 exon	9	Likely pathogenic	PM2, PM5, PP3, PM1, PP4	SIFT, SIFT4G, Polyphen2_HDIV, Polyphen2_HVAR, FATHMM, PROVEAN, DEOGEN2
4.1	4	m.	c.1547G>A	p.Cys516Tyr	10 exon	13	Pathogenic	PM1, PM5, PP5, PM2, PP3	MutPred, DEOGEN2, M-CAP, Mutation assessor, MVP, FATHMM, FATHMM-MKL, PROVEAN, EIGEN, EIGEN PC, LIST-S2, FATHMM-XF, PrimateAI, SIFT, SIFT4G
4.2	4	m.	c.1547G>A	p.Cys516Tyr	10 exon	13	Pathogenic	PM2, PP3, PP1, PP4, PM5, PM1	MutPred, DEOGEN2, M-CAP, Mutation assessor, MVP, FATHMM, FATHMM-MKL, PROVEAN, EIGEN, EIGEN PC, LIST-S2, FATHMM-XF, PrimateAI, SIFT, SIFT4G

## Data Availability

Not applicable.

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
