# Peer review of "Four Novel Disease-Causing Variants in the NOTCH3 Gene in Russian Patients with CADASIL"

_genes, 2023, doi:10.3390/genes14091715_

Round 1
Reviewer 1 Report
The manuscript of Bostanov et.al is written in an understandable way and I have no objections to the structure itself. Let the authors check because not everywhere the names of genes are written in italics.
My biggest message, because it is not an allegation, is the fact that only in one case is there a clear phenotype-genotype correlation confirmed in family members. I know that in the rest, as you can see in the figures, it was impossible, but this should be clearly noted in the text. Without such an important component of confirming genotype causality, we must take inferences very carefully.
In addition, I lack a broader description of where these families were examined, because it was rather the result of a longer work of a team of clinicians. Because such cases do not happen every day and maybe it would be good to outline at least in one sentence how long and from how many other not so unambiguous cases these just selected ones emerged.
Author Response
Dear Editor and Reviewers, thank you for your careful review of our article. We have tried to make changes in accordance with your comments.
Please see the attachment
Response to Reviewer 1 Comments
Point 1: The manuscript of Bostanov et.al is written in an understandable way and I have no objections to the structure itself. Let the authors check because not everywhere the names of genes are written in italics.
Response 1: The correctness of the gene names has been verified.
Point 2: My biggest message, because it is not an allegation, is the fact that only in one case is there a clear phenotype-genotype correlation confirmed in family members. I know that in the rest, as you can see in the figures, it was impossible, but this should be clearly noted in the text. Without such an important component of confirming genotype causality, we must take inferences very carefully.
Response 2: Changes have been made according to your suggestions.
Point 3: In addition, I lack a broader description of where these families were examined, because it was rather the result of a longer work of a team of clinicians. Because such cases do not happen every day and maybe it would be good to outline at least in one sentence how long and from how many other not so unambiguous cases these just selected ones emerged.
Response 3: Most patients were examined by neurologists in the inpatient setting and at the Multiple Sclerosis Center where they lived. As the data are archival, it was difficult to obtain more detailed information about the specialist team. We tried to summarize these data in the text. Between 2007 and 2022, 72 cases of CADASIL were diagnosed at the Research Centre for Medical Genetics. Among them, 4 previously undescribed variants were identified. In the text, this sentence was added to the "Background".

Reviewer 2 Report
The authors present in their small case series new novel disease-causing variants related to CADASIL. I have minor suggestions to the manuscript at this point. The first one is that authors should review their Table 1 in the manuscript, regarding the column about evidence of pathogenicity. For some of the variants evaluated by the authors, the ACMG criteria which were used do not follow the most recent revised criteria published by the ACMG (the analysis was almost done following the rules on 2015's criteria). One additional interesting point and suggestion is to authors present the most important data related to the "in-silico" predictions tools for each of the four new variants.
Author Response
Dear Editor and Reviewers, thank you for your careful review of our article. We have tried to make changes in accordance with your comments.
Please see the attachment
Response to Reviewer 2 Comments
Point 1 The authors present in their small case series new novel disease-causing variants related to CADASIL. I have minor suggestions to the manuscript at this point. The first one is that authors should review their Table 1 in the manuscript, regarding the column about evidence of pathogenicity. For some of the variants evaluated by the authors, the ACMG criteria which were used do not follow the most recent revised criteria published by the ACMG (the analysis was almost done following the rules on 2015's criteria).
Response 1: We revised Table 1 in the manuscript and, in the column on evidence of pathogenicity, made changes according to the latest revised criteria published by the ACMG.
Point 2 One additional interesting point and suggestion is to authors present the most important data related to the "in-silico" predictions tools for each of the four new variants.
Response 2: We have entered the data related to the in-silico forecasting tools for each of the four new variants.
